# Identification of Compounds Preventing *A. fumigatus* Biofilm Formation by Inhibition of the Galactosaminogalactan Deacetylase Agd3

**DOI:** 10.3390/ijms24031851

**Published:** 2023-01-17

**Authors:** Carla I. I. Seegers, Danielle J. Lee, Patricia Zarnovican, Susanne H. Kirsch, Rolf Müller, Thomas Haselhorst, Françoise H. Routier

**Affiliations:** 1Institute for Clinical Biochemistry, OE4340, Hannover Medical School, Carl-Neuberg-Strasse 1, 30625 Hannover, Germany; 2Institute for Glycomics, Griffith University, Gold Coast Campus, Queensland 4222, Australia; 3Department of Microbial Natural Products, Helmholtz Institute for Pharmaceutical Research Saarland (HIPS), Helmholtz Center for Infection Research, Saarland University Campus, 66123 Saarbrücken, Germany; 4German Center for Infection Research (DZIF), Partner Site Hannover-Braunschweig, 38124 Braunschweig, Germany; 5Department of Pharmacy, Saarland University, 66123 Saarbrücken, Germany

**Keywords:** polysaccharide, deacetylase, biofilm, inhibitor, *Aspergillus fumigatus*

## Abstract

The opportunistic fungus *Aspergillus fumigatus* causes a set of diseases ranging from allergy to lethal invasive mycosis. Within the human airways, *A. fumigatus* is embedded in a biofilm that forms not only a barrier against the host immune defense system, but also creates a physical barrier protecting the fungi from chemicals such as antifungal drugs. Novel therapeutic strategies aim at combining drugs that inhibit biofilm synthesis or disrupt existing biofilm with classical antimicrobials. One of the major constituents of *A. fumigatus* biofilm is the polysaccharide galactosaminogalactan (GAG) composed of α1,4-linked N-acetylgalactosamine, galactosamine, and galactose residues. GAG is synthesized on the cytosolic face of the plasma membrane and is extruded in the extracellular space, where it is partially deacetylated. The deacetylase Agd3 that mediates this last step is essential for the biofilm formation and full virulence of the fungus. In this work, a previously described enzyme-linked lectin assay, based on the adhesion of deacetylated GAG to negatively charged plates and quantification with biotinylated soybean agglutinin was adapted to screen microbial natural compounds, as well as compounds identified in in silico screening of drug libraries. Actinomycin X2, actinomycin D, rifaximin, and imatinib were shown to inhibit Agd3 activity in vitro. At a concentration of 100 µM, actinomycin D and imatinib showed a clear reduction in the biofilm biomass without affecting the fungal growth. Finally, imatinib reduced the virulence of *A. fumigatus* in a *Galleria mellonella* infection model in an Agd3-dependent manner.

## 1. Introduction

Fungal infections affect billions of people worldwide and cause millions of deaths annually, following severe respiratory illnesses and infections of the bloodstream. One of the most lethal fungi is *Aspergillus fumigatus*, which causes several diseases including allergic bronchopulmonary aspergillosis (ABPA), chronic pulmonary aspergillosis (CPA), and invasive aspergillosis (IA) [1]. Patients suffering from ABPA and CPA develop poorly controlled asthma and recurrent pulmonary infections which leads to complications of many other respiratory conditions, such as cystic fibrosis [2,3]. In contrast, IA is not restricted to the lung as it can spread to the other organs via the bloodstream [4,5]. It mostly occurs in patients with severe immune deficiency (e.g., patients with chemotherapy-induced neutropenia or transplant recipients) or severe respiratory infections (influenza or COVID-19) [6]. More than 80% of all IA are caused by *A. fumigatus* and are associated with a high mortality rate of up to 30–50% [7,8,9]. Along with the immunocompromised population, the incidence of IA is constantly growing, but the therapy remains problematic.

*Aspergillus fumigatus* grows as a typical biofilm with hyphae covered by an extracellular matrix composed of polysaccharides, proteins, extracellular DNA, lipids, and polyols [10]. The extracellular matrix of *Aspergillus* biofilms formed in vivo contains large amounts of the polysaccharides galactomannan and galactosaminogalactan (GAG) [11]. These two polysaccharides are both secreted and associated with the surface of the hyphae and thus form the interface between the fungus and the host. Galactomannan is synthesized along the secretory pathway and is essential for the growth and virulence of *A. fumigatus* [12,13,14,15,16,17,18]. GAG is a linear heteropolymer of α-1,4-linked galactose (Gal), N-acetylgalactosamine (GalNAc), and galactosamine (GalN) and is synthesized at the plasma membrane by enzymes encoded by a cluster of five co-regulated genes, conserved in filamentous fungi [19]. The cluster encodes a UDP-glucose epimerase (Uge3) that provides both UDP-Gal and UDP-GalNAc for the putative polymerase Gt3b [20]. It also encodes the endoglycosidases Sph3 and Ega3 specific for α-1,4-GalNAc and α-1,4-GalN, respectively [21,22]. Finally, Agd3 partially deacetylates α-GalNAc residues of GAG and renders the polysaccharide polycationic and adherent to negatively charged surfaces. Accordingly, the deletion of *agd3* leads to the loss of adhesion of GAG to the fungal hyphae and to the epithelial cells of the airways. The fully acetylated GAG is non-functional and sheds into the culture medium [19]. Deacetylation of GAG is thus required for the biofilm formation and therefore full virulence [19,23]. In addition to mediating the adhesion of hyphae to host cells and other surfaces, GAG protects the fungus from the host immune system and from antifungal chemotherapy [24]. Notably, GAG is involved in the masking of β-1,3-glucans from immune recognition and helps to resist neutrophil extracellular traps [23,25].

Given its key role in biofilm formation and pathogenicity as well as its extracellular localization, the fungal deacetylase Agd3 represents an attractive target for the development of novel therapeutics that would attenuate virulence and increase susceptibility to antifungals. In this project, we used two different approaches for the identification of Agd3 inhibitors and demonstrated their value in a *Galleria mellonella* infection model.

## 2. Results

### 2.1. Establishment of an Agd3 Enzymatic Activity Assay

Agd3 lacking the twin-arginine translocation (TAT) signal sequence and part of the N-terminal serine-rich region was expressed in the yeast *Pichia pastoris* X-33 and purified by nickel affinity chromatography and gel filtration. To display the enzymatic activity of the purified Agd3 protein, assays using the common deacetylase substrates 4-nitrophenol acetate or 4-methylumbelliferyl acetate [26,27,28] were performed but no activity was detected. Similarly, attempts were made to reveal the Agd3 activity using GalNAc-α-*p*-nitrophenol as a substrate and with the detection of primary amine by fluorescamine, which was adapted from Blair et al. (2015) [29]. However, these attempts were unsuccessful.

According to Lee et al., the deletion of *agd3* in *A. fumigatus* Af293 led to the synthesis of fully acetylated GAG that was non-adherent and was released in the supernatant [19]. In order to obtain this natural substrate of Agd3, we deleted the *agd3* gene by homologous directed repair in the AfS35 strain, which is a derivative of the clinical strain D141 (Figure 1A) [30]. Gene deletion was confirmed by PCR (Figure 1B). The resulting Δ*agd3* strain did not display any notable growth phenotype in a drop dilution assay (Figure 1C). As expected, crystal violet staining highlighted the absence of adhesive biofilm at the surface of the hyphae (Figure 1D). Biofilm formation could however be restored by the addition of 1 nM of purified Agd3, which indirectly demonstrates the enzymatic activity of the recombinant Agd3. Enzymatic activity of the purified Agd3 was further confirmed in an enzyme-linked lectin assay (ELLA) using a culture supernatant of Δ*agd3* containing fully acetylated GAG as shown below [31]. 

### 2.2. Screening of Microbial Compounds Revealed Inhibition of Agd3 by Actinomycin D

In the first approach, we tested the influence of 262 natural products (a subset of the German Center for Infection Research (DZIF) Natural Compound Library, containing 88 compounds derived from fungi and 174 natural products from bacteria and fungi from BioViotica Naturstoffe GmbH, Dransfeld, Germany) with high structural and functional diversity on the Agd3 enzymatic activity. Firstly, 50 µL culture medium of the Δ*agd3* strain containing acetylated GAG was incubated with 1 nM recombinant Agd3 pre-incubated with 1 µM of a natural product. After deacetylation, the GAG bound to the 96-well plate was detected using soybean agglutinin (SBA)-biotin and streptavidin-horseradish peroxidase (HRP). As a final readout, the absorbance of the HRP substrate tetramethylbenzidine (TMB) was measured (Figure 2). The obtained absorbance values were compared to the samples without inhibitors, which are considered 100% activity and are indicated by a red reference line in Figure 2A. Samples without recombinant Agd3 or with 1 nM Agd3 pre-incubated with the chelator 2,6-Pyridinedicarboxylic acid (DPA) were used as negative controls and were considered as 100% inhibition (Figure 2A, circled in red). The screening revealed two compounds that inhibited Agd3: actinomycin X2 (Figure 2A, compound **4**) and actinomycin D (Figure 2A, compound **52**).

We then docked actinomycin D into the active site of Agd3 using the X-ray crystal structure of Agd3 and AutoDock Vina, implemented into the YASARA software suite (Yet Another Scientific Artificial Reality Application, Version 19.12.14) [32]. A rectangular box with a dimension of 40 Å × 40 Å × 40 Å (x, y, and z) was centered around the amino acid His-514, which is part of the active center of the protein [31]. The 3D structure of actinomycin D was obtained from PubChem [33] as an sdf file, which was then imported into Datawarrior (Version 5.2.1) [34] to obtain 3D coordinates. The AutoDock Vina was performed for 12 docking orientations of actinomycin D, which were then ranked by binding energy into 16 distinct clusters. All clusters were similar, and actinomycin D occupied the binding site of Agd3. Figure 3A,B shows actinomycin D bound to Agd3 in a surface model (A) and a more detailed representation (B) with hydrophobic interaction displayed as green lines. From this docked structure, it is apparent that almost the entire binding pocket of Agd3 is blocked by actinomycin D, which could explain the inhibition of Agd3 activity.

We subsequently tested the effect of actinomycin D on the biofilm formation of *A. fumigatus* S35 using crystal violet staining (Figure 2B). The addition of 10 µM actinomycin D was shown to have no visible influence on the biofilm, while at a concentration of 100 µM, it destabilized the biofilm. Quantification of the crystal violet staining showed a decrease in the biofilm biomass by approximately 35% with the addition of 100 µM actinomycin D (Figure 2C). We additionally analyzed the growth of *A. fumigatus* in liquid culture in the presence of various concentrations of actinomycin D ranging from 0.4 to 200 µM. The fungal growth was not affected by the addition of actinomycin D and therefore excludes the possible fungistatic or fungicidal activity. 

Actinomycin D (isolated from *Streptomyces parvulus*) is an inhibitor of transcription used as chemotherapy for highly aggressive cancers and is known for cytotoxic effects. To assess if the effect of actinomycin D on the biofilm of *A. fumigatus* could be shown in a *Galleria mellonella* infection model, we injected a group of 20 larvae with actinomycin D. The viability of the larvae was compared to an untreated group as well as a group injected with the buffer only. Based on the doses of antifungals used for the treatment of *A. fumigatus* infected *G. mellonella* larvae [35], we chose to inject the larvae with 1.5 nmol actinomycin D (corresponding to ~1.9 µg). Unfortunately, all larvae injected with actinomycin D died within 24 h while the survival rate was at least 95% in the control groups. This control experiment highlighted the toxicity of actinomycin D and precluded performing further tests in this model.

### 2.3. Structure-Based Virtual Screen Display Inhibition of Agd3 by Imatinib and Rifaximin

In an additional approach, we used molecular docking to screen 18,673 ligands from multiple compound libraries from chemical vendors consisting of drugs, dyes, and other therapeutic molecules. Again, a rectangular box with a dimension of 40 Å by 40 Å by 40 Å (x, y, and z) was centered around His-514 within the active center of Agd3 [31]. Each 3D structure contained in the library was docked using AutoDock Vina implemented in the YASARA software suite following our published in silico screening approach [36]. Hits were scored using the theoretical dissociation constant (K_D_) and the top hits with a theoretical K_D_ of less than 200 nM were visualized and purchased. The ability of these compounds to inhibit Agd3 activity was then tested in a GAG-ELLA assay as described above (Figure 4A). Imatinib and rifaximin showed a clear inhibitory effect at a concentration of 100 µM in the GAG-ELLA assay (Figure 4A).

Figure 3C,D shows the docked structure of imatinib bound to Agd3 in a surface model (A) and a more detailed representation (B) with hydrophobic interaction represented by the green lines and ion–pi interactions in red. As seen in this docked structure, imatinib occupies almost the entire binding pocket of Agd3. The interactions with the amino acids in the imatinib-Agd3 model are different compared to the actinomycin D-Agd3 model (Figure 3A,B), due to the inherently different sizes of these two inhibitors. Interestingly, imatinib interacts with the catalytic base His668.

Crystal violet staining revealed that the addition of 100 µM rifaximin to the culture medium of AfS35 did not compromise the biofilm formation. Due to its inability to disrupt the biofilm and its limited solubility in aqueous systems, rifaximin was not analyzed further. In contrast, the wild-type AfS35 strain produced a fragile and thin biofilm (Figure 4B) in the presence of 100 µM imatinib showing a reduction in the formation of approximately 50% (Figure 4C). Further, in the presence of 1 mM, the biofilm was eliminated (Figure 4B). However, the growth of *A. fumigatus* was not affected in the liquid culture with the addition of imatinib ranging from 2 µM to 1 mM. 

### 2.4. Imatinib Reduced A. fumigatus Virulence in a Galleria Mellonella Infection Model

To strengthen these results, we analyzed the effect of imatinib on *Aspergillus fumigatus* infection in *Galleria mellonella* larvae (Figure 4D). When the larvae were injected with 10^6^ spores of AfS35 strain (wt), the mortality was close to 100% within 48 h. In contrast, larvae infected with 10^6^ Δ*agd3* spores displayed significantly lower mortality. This is in agreement with the reduced virulence of *A. fumigatus* Af293 deficient in Agd3 [19]. An intermediate 48 h mortality rate of 86% was obtained when 1.5 nmol imatinib (corresponding to 0.75 µg/larvae) were injected with AfS35 spores. At this dose, imatinib alone did not affect the larval viability. Importantly, the reduction in virulence induced by imatinib was not observed with the Δ*agd3* strain and therefore excludes an Agd3-independent effect of imatinib. 

## 3. Discussion

Novel combinatorial therapies aims to prevent the initiation of a biofilm or aims to disrupt the existing biofilms leading to simultaneously targeting and killing the pathogenic microorganisms [37]. Previously, the glycosylhydrolase Sph3, which cleaves galactosaminogalactan (GAG) was shown to disrupt the biofilm and thereby increase the sensitivity of *A. fumigatus* to the different classes of antifungals in vitro and enhance the efficiency of posaconazole in a mouse model of IA [38,39]. Here, we demonstrate that compounds targeting Agd3 activity may also reduce the *A. fumigatus* biofilm. 

Lee and collaborators have previously proven that the synthesis of fully acetylated GAG resulting from *agd3* deletion led to the disruption of the biofilm and therefore the attenuation of the strain Af293 [19]. Recently, it has been reported that this laboratory strain might be infected by a mycovirus that decreases fungal fitness [40]. Here, the deletion of *agd3* in the *A. fumigatus* strain S35, a derivative of the clinical strain D141 [30], confirmed that the critical roles of the deacetylase in GAG adhesiveness, biofilm formation, and fungal virulence are not strain specific. As previously shown, subnanomolar concentrations of recombinant Agd3 added to the culture medium of the Agd3 deficient strain (Δ*agd3*) were sufficient to restore the biofilm [31]. This result suggests that a high level of Agd3 inhibition is necessary to prevent biofilm formation in vivo. 

To develop an enzymatic assay for Agd3, we first tested potential synthetic substrates. Polysaccharide deacetylases, including *Pseudomonas aeruginosa* PelA, commonly use *p*-nitrophenyl acetate in vitro [41]. Although Agd3 and PelA act on similar α1,4-linked GalNAc containing heteropolymer in vivo [42,43], *p*-nitrophenyl acetate was not a substrate of Agd3. Similarly, GalNAc-α-*p*-nitrophenol did not allow the detection of Agd3 activity. The absence of enzymatic activity on these small synthetic substrates is in agreement with the preference of the enzyme for α1,4-GalNAc oligosaccharides longer than 13 residues and its poor activity on 7-mers [31]. Consequently, we used acetylated GAG secreted by the Δ*agd3* strain as a substrate. Using an enzyme-linked lectin assay, we assessed the effect of a small set of bacterial or fungal natural compounds on the deacetylase activity of Agd3 and identified the transcription inhibitor actinomycin D and its analog actinomycin X2 as Agd3 inhibitors. However, at a concentration of 100 µM, actinomycin D could only partially disrupt the biofilm formation and injection of 1.5 nmol was found toxic to *Galleria mellonella* larvae. In comparison, actinomycin D suppresses the biofilm formation of *Staphylococcus epidermidis* at a concentration of 3 µg/mL (2.3 µM) by decreasing the expression of the polysaccharide deacetylase IcaB and thereby affecting the composition and adhesion of the biofilm [44]. Since the amounts required to significantly reduce the *A. fumigatus* biofilm formation are toxic, actinomycin D was not analyzed further. 

Additional potential Agd3 inhibitors were then identified by in silico screening of compound libraries against the crystal structure of Agd3 [31]. Compounds with low theoretical K_D_ values were tested on their ability to inhibit the deacetylase activity using enzyme-linked lectin assay. This led to the identification of an additional Agd3 inhibitor, imatinib, which is a selective inhibitor of ABL tyrosine kinases used for the treatment of cancer [45]. In vitro, 100 µM imatinib significantly reduced the biofilm biomass in the crystal violet assay. Moreover, the injection of 1.5 nmol imatinib to *Galleria mellonella* larvae (corresponding to 0.75 µg/larvae or ~3 mg/kg) was well tolerated and improved the viability of larvae infected with *A. fumigatus* S35 strain. However, Agd3 inhibition was likely incomplete in this case since the survival rate was higher in the larvae infected with the Δ*agd3* strain than in larvae infected with the parental S35 strain and treated with imatinib. In patients with chronic myeloid leukemia, an average dose of 400 mg imatinib is administered orally once a day with few adverse effects and leads to a steady state plasma concentration of 2.6 +/− 0.8 µg/mL at peak [46]. Further studies are necessary to evaluate if non-toxic doses of imatinib or derivatives of imatinib would provide a benefit for the treatment of drug-resistant strains of *A. fumigatus*. 

Orthologous *agd3* genes are present in fungi as well as in some Gram-negative and Gram-positive bacteria. The conservation of catalytic residues and localization of the fungal *agd3* orthologues in clusters resembling the *A. fumigatus* GAG gene cluster suggests a similar function in biofilm formation [31]. The bacterial orthologues are more distant but also contain the principal catalytic motifs and are often situated in operons encoding a glycosyltransferase and an endo-α1,4-galactosaminidase Ega3 homolog [22,31]. Inhibitors of Agd3 might be useful to assess the role of these orthologues in biofilm formation and adherence. 

*Aspergillus fumigatus* often coexist with the Gram-negative bacterium *Pseudomonas aeruginosa* in the airways of cystic fibrosis patients with deleterious consequences on pulmonary functions [3,47]. Both pathogens are protected by biofilm formation, which represents an additional challenge for therapy. *P. aeruginosa* synthesizes the polysaccharide pellicle (PEL), a polymer of α1,4-linked N-acetylgalactosamine (GalNAc) and N-acetylglucosamine (GlcNAc) that resembles *A. fumigatus* galactosaminogalactan (GAG) [42]. As Agd3, *P. aeruginosa* deacetylase pelA partially deacetylates the PEL polysaccharide and is essential for the biofilm formation and full virulence [48]. Recent studies have shown that GAG can be incorporated into *P. aeruginosa* biofilm in vitro. The presence of deacetylated GAG is associated with an increase in adherent bacterial biofilm formation and a decrease in bacterial killing by antibiotics [39]. Inhibitors of Agd3 might thus benefit the treatment of cystic fibrosis patients co-infected by *Aspergillus* and *Pseudomonas*. 

## 4. Materials and Methods

### 4.1. Strains and Growth Conditions of A. fumigatus

In this study, *A. fumigatus S35*, a derivate of the clinical isolate D141 deficient in non-homologous end-joining was used [30]. The fungal strain and derivatives were grown in *Aspergillus* minimal medium (AMM) or AMM supplemented with 2% agar.

### 4.2. Expression and Purification of Agd3 from Pichia Pastoris X-33

The region of the *agd3* gene encoding amino acids 109–801 was amplified from cDNA of *A. fumigatus S35* using the primers CS27 and CS29 (Table 1) and cloned into the vector pPICZαA (Invitrogen) by golden gate cloning using the restriction sides EcoRI-HF and XbaI (NEB). The SacI linearized vector was transformed in *Pichia Pastoris X-33* via electroporation and clones were selected with Zeocin^TM^ according to the manufacturer’s instructions. Expression of Agd3_109–801_ was confirmed by Penta·His Antibody Western Blot staining (Quiagen). A selected clone was grown in 500 mL buffered glycerol complex medium until OD_600_ = 1 and protein expression was induced with 0.5% methanol for 24 h. The filtered supernatant was loaded on a 5 mL HisTrap™ HP column. The column was washed with 20 mM HEPES pH 7.5 + 200 mM NaCl and protein was eluted with a gradient of 70% 20 mM HEPES, pH 7.5 + 200 mM NaCl + 500 mM Imidazole over 10 min. The protein was purified using a Superdex 200 10/300 GL resulting in a yield of 6–9 mg per 500 mL culture.

### 4.3. Generation of ∆agd3 Knockout

CRISPR/Cas9 plasmids and marker-free repair DNA fragments were generated according to Seegers et al. (2022) [49]. Briefly, protospacer sequences were selected via the online tool CHOP-CHOP [50]. The tRNA^Pro1^ promoter, tRNA, and crRNA were amplified from plasmid pTLL108.1 [51,52] and fused to either one protospacer sequence using primers FR294 and FR297 to generate sgRNA1 targeting the beginning of the gene or FR294 and FR298 for sgRNA2 targeting the end of the gene (Table 1). The tracrRNA and tRNA terminator sequences were amplified from plasmid pTLL109.2 [52] and both fragments were cloned into the vector pFC333 (Addgene plasmid #87844) [53]. The final constructs were sequenced. To generate the marker-free repair DNA, 1 kb fragments corresponding to the sequence upstream and downstream of *agd3* were amplified from genomic DNA using primers FR299/FR300 (5′-UTR of the gene) and FR301/FR302 (3′-UTR of the gene) (Table 1). These fragments were then fused by PCR using the 5′-UTR forward primer and 3′-UTR reverse primer. The amounts of 1 µg of each CRISPR/Cas9 plasmid and 2 µg repair DNA fragment were transformed into *A. fumigatus S35* by PEG-mediated fusion of protoplasts adapted from Punt et al. [54]. Transformants were selected on AMM containing 2% Agar, 1.2 M Sorbitol, and 30 µg/mL Phleomycin and incubated for 24 h at RT. The plates were transferred to 37 °C in high humidity for 2–4 days until first colonies appeared. The clones were further singled out on AMM plates without antibiotics to allow loss of the CRISPR/Cas9 plasmid. Gene deletion was confirmed by PCR using the primers FR299 and FR302 and CS14/CS37.

### 4.4. A. fumigatus Growth

Series of ten-fold dilutions of spores of *AfS35* and ∆*agd3* were performed in PBS to obtain solutions of 2 × 10^8^ to 2 × 10^4^ spores/mL and 5 µL (corresponding to 10^6^ to 10^2^ spores) were spotted onto AMM agar plate. The plates were incubated at 37 °C in high humidity and imaged after 20 h.

To determine the growth of *A. fumigatus S35* in presence of actinomycin D (BioViotica Naturstoffe GmbH, Dransfeld, Germany) and imatinib (Santa Cruz), we performed 1:2 serial dilutions in RPMI 1640 medium with 2% glucose. In parallel, a fresh spore solution was adjusted to 4 × 10^6^ spores/mL in RPMI medium with 2% glucose. Then, 100 µL spore solution was added to 100 µL of each inhibitor dilution in a CELLSTAR 96 well cell culture plate (Greiner) to obtain a final inhibitor concentration of 1 mM–1.95 µM for imatinib or 200–0.39 µM for actinomycin D. As a reference, spore solution was inoculated with 100 µL RPMI medium with 2% glucose containing 0.5% DMSO without the inhibitor. The plate was incubated at 37 °C in high humidity and turbidity was measured via absorbance at 530 nm after 24 h and 48 h.

### 4.5. Enzyme-Linked Lectin Activity Assay for Agd3

To test the activity of purified Agd3, a GAG ELLA was performed according to Bamford et al. (2020) [31]. Briefly, 100 mL AMM were inoculated with 10^7^ spores/mL of ∆*agd3* and incubated at 37 °C for 18 h at 225 rpm. Fungal debris were removed by centrifugation at 1500× *g* for 10 min and 50 µL of the supernatant was transferred into each well of a Immulon 4 HBX 96 well microtiter plate. An amount of 50 µL of a 1 nM Agd3 solution was added to each well and the plate was incubated for 1h at RT. Controls without Agd3 or with 1 nM Agd3 pre-incubated with 5 mM of the chelator dipicolinic acid (DPA) for 30 min at RT were performed. Following incubation, the plate was washed three times with PBS + 0.05% Tween-20 (PBST) and blocked with 0.05 M Tris-HCl (pH 7.5) + 0.15 M NaCl containing 1 mM MgCl_2_, 1 mM CaCl_2_ and 1 mM MnCl_2_ (blocking buffer) for 1 h at RT. The samples were then incubated with 50 µL of 5 µg/mL SBA-Biotin in blocking buffer for 1 h and further processed with streptavidin-HRP (Invitrogen), diluted 1:50,000 in the blocking buffer for 1 h at RT. The plate was washed 3 times with PBST after each step. The ELLA was developed by adding 50 µL of 100 mM sodium acetate/100 mM citric acid buffer pH 4.9 containing 10% 1 mg/mL TMB solution dissolved in DMSO and 0.03% H_2_O_2_ before incubation at RT for 5 min. Subsequently the reaction was stopped by adding 20 µL of 1 M H_2_SO_4_ and absorbance was detected at 450 nm. To test for potential inhibitors, 1 µL of a 100 µM compound solution in DMSO was added to the wells and incubated with Agd3 for 30 min at RT. A control with 1 µL DMSO was performed to ensure that it did not affect the Agd3 activity.

### 4.6. Molecular Docking of Agd3

A molecular docking screen was carried out following published procedures [34]. Briefly, the three-dimensional structure of Agd3 was obtained from the RCSB protein databank (6NWZ, [31]). A rectangular box with a dimension of 40 Å by 40 Å by 40 Å (x, y, and z) was fitted around the amino acid His514 which is in the active center of the protein [31]. We have chosen a subset of ligands with high diversity resulting in 18,673 ligands (Charitee Super Drugs, 305 compounds; Approved Drugs, 3149 compounds; Drugbank, 1909 compounds; eDrugs, 3104 compounds; Prestwick Off-Patents drugs, 2062 compounds; and Otava, 8144 compounds) and edited for screening as described in Day et al. (2021) [36]. After performing the macro dock screening, the hits were further refined by target and ligand optimization as implemented by Day et al. (2021) [36].

### 4.7. Biofilm Staining of A. fumigatus Strains by Crystal Violet

To determine the biofilm biomass, 50 µL of AMM medium was inoculated with 5 × 10^6^ spores/mL of *A. fumigatus* S35 in each well of a CELLSTAR 96-well plate (Greiner) and was incubated at 37 °C for 24 h with or without the addition of 1, 10, or 100 µM rifaximin (LGC Labor GmbH), imatinib (Santa Cruz) or actinomycin D (BioViotica Naturstoffe GmbH, Dransfeld, Germany). As controls, wells containing only AMM were measured. The wells were washed twice with H_2_O and further stained with 0.1% crystal violet in H_2_O. After incubation, the crystal violet solution was removed, and the remaining biofilm was washed three times with H_2_O. For quantification purposes, the wells were destained with 150 µL of 95% ethanol for 10 min and the absorbance was measured at 600 nm. For imaging, the stained biofilm was air-dried and imaged using a mosaic image calculation on the Axio Observer Z1 microscopic system with a 5x objective. Experiments with Δ*agd3* were performed with and without the addition of 1 nM purified Agd3.

### 4.8. Virulence Studies in Galleria Mellonella Larvae

Spores of *Af*ΔAgd3 and *Af*S35 were freshly harvested and diluted to a concentration of 6.67 × 10^7^ spores/mL in sterile PBS. Groups of 20 larvae (~250 mg) were infected in the last right proleg with 15 µL containing 10^6^ spores with or without the inhibitor. As controls, there were untreated larvae, larvae inoculated with sterile PBS, or injected with the inhibitor only. After injection, larvae were incubated at 37 °C and the viability was monitored after 24 and 48 h.

### 4.9. Statistical Analysis

All graphs were generated and statistically analyzed using the software GraphPad Prism (version 5.01, GraphPad, San Diego, CA, USA). Significant differences between values were calculated using a one-way analysis of variances (ANOVA) with a Bonferroni’s multiple comparison test and indicated by * (*p* < 0.05), ** (*p* < 0.01), or *** (*p* < 0.001), whereas non-significant differences are indicated by ns.

## Figures and Tables

**Figure 1 ijms-24-01851-f001:**
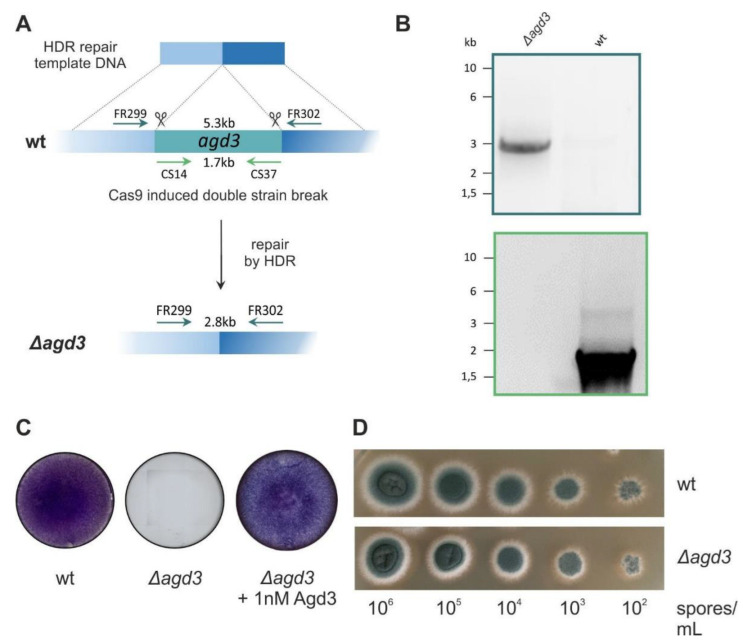
Deletion of *agd3* in *A. fumigatus* S35 strain. (**A**) Scheme of homologous directed repair strategy used for *agd3* excision. (**B**) Validation of gene deletion by PCR reactions. The fragment of 2.8 kb obtained with primers FR299 and FR302 and absence of *agd3* gene amplification with primers CS14 and CS37 confirm gene deletion. (**C**) The 34 h old biofilms were visualized by crystal violet staining. Formation of an adherent biofilm was not observed in the Δ*agd3* strain compared to wt (*Af*S35) but was restored with the addition of 1 nM purified Agd3 to the growth medium. (**D**) Growth of Δ*agd3* and parental AfS35 strain (wt) was analyzed in a drop dilution assay (10^6^ to 10^2^ spores) on AMM agar plate at 37 °C for 20 h.

**Figure 2 ijms-24-01851-f002:**
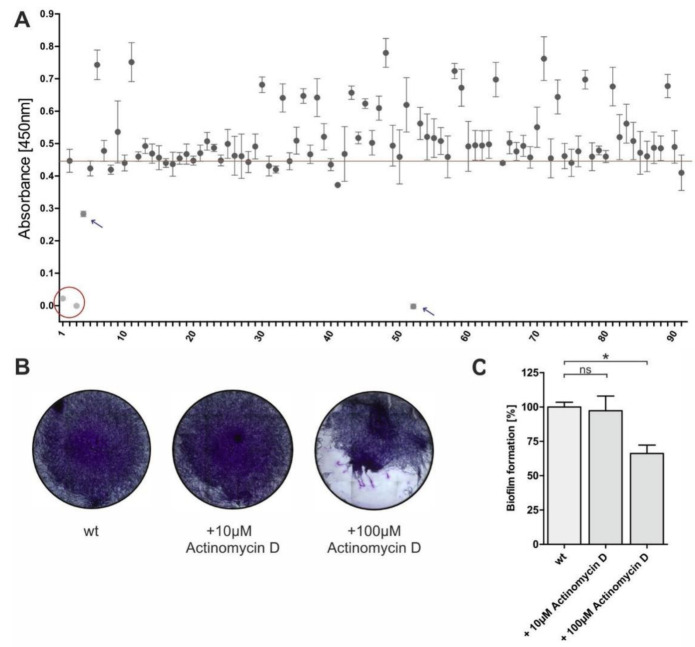
Effect of microbial natural compounds on Agd3 enzymatic activity. (**A**) Microbial natural compounds from bacteria and fungi were screened for their ability to inhibit Agd3 activity in a GAG ELLA assay. Results from a plate containing 88 compounds are displayed. The red line indicates 100% activity determined by measuring the Agd3 activity in absence of additional compound. Samples without Agd3 enzyme (position 1, circled in red) or with Agd3 incubated with 5 mM dipicolinic acid (position 2, circled in red) were used as negative controls. Compounds **4** and **52** marked with a blue arrow inhibited Agd3 activity (n = 3). (**B**) Actinomycin D (Compound **52**) showed the inhibition of the biofilm formation in a crystal violet staining. (**C**) Quantification of the biofilm stain via absorbance measurement (n = 3). Statistical analysis was run by one-way repeated measures ANOVA and Bonferroni’s multiple comparison test. * (*p* < 0.05): ns: non-significant.

**Figure 3 ijms-24-01851-f003:**
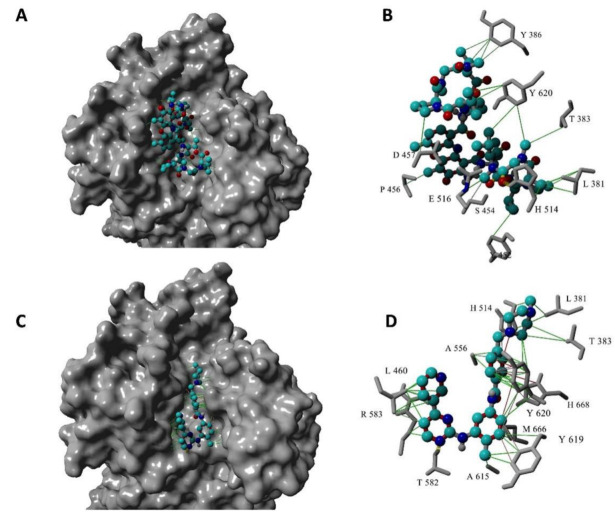
Docking of actinomycin D and imatinib in Agd3. (**A**) Actinomycin D occupies Agd3 catalytic center. (**B**) Interactions of actinomycin D with Agd3 amino acids. (**C**) Imatinib in Agd3 catalytic center. (**D**) Interactions of imatinib with Agd3 amino acids including the catalytic residues His514 and His668.

**Figure 4 ijms-24-01851-f004:**
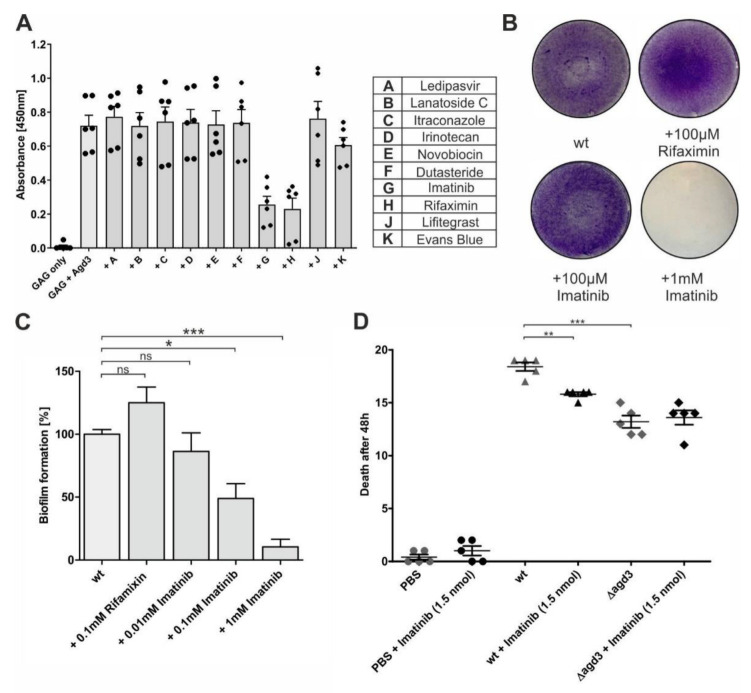
Inhibition of Agd3 activity by imatinib and rifaximin. (**A**) Ten potential inhibitors selected by molecular docking screening were tested at a 100 μM concentration in a GAG ELLA assay (n = 3). (**B**) Effect of 100 µM imatinib or rifaximin on AfS35 biofilm visualized by crystal violet staining. (**C**) Quantification of the biofilm staining via absorbance measurement (n = 3). (**D**) Testing of imatinib in a *Galleria mellonella* infection model. Groups of 20 larvae were injected with 15 µL containing PBS, PBS + 1.5 nmol imatinib, 10^6^ spores of *Af*S35 strain (wt), 10^6^ spores of *Af*S35 strain + 1.5 nmol imatinib, 10^6^ spores of Δ*agd3*, or 10^6^ spores of Δ*agd3* + 1.5 nmol imatinib. The experiment was performed with n = 5 from four different larvae batches and statistical analysis was run by one-way repeated measures ANOVA and Bonferroni’s multiple comparison test. * (*p* < 0.05), ** (*p* < 0.01) or *** (*p* < 0.001), ns non-significant.

**Table 1 ijms-24-01851-t001:** Primers used in this study.

Name	Primer Sequence *	Description
CS27	GTAC**GGTCTCGAATTC**TCGATTGGCCTGTCCTCC	Forward *agd3*_109–801_
CS29	GCTA**TCTAGA**TGCAAAGCAATAGGAGTGGAAAGCG	Reverse *agd3*_109–801_
FR294	GACA**GGTCTCAGATCT**ACTCCGCCGAACGTACTG	Forward CRISPR/CAS9
FR297	GACA**GGTCTCCAAACT**GGTGAAATACGGCCACATGGACGAGCTTACTCGTTTCGTCCTC	Reverse CRISPR/CAS9
FR298	GACA**GGTCTCCAAACT**CGTTACATCTGTGCCTCCAGACGAGCTTACTCGTTTCGTCCTC	Reverse CRISPR/CAS9
FR299	CATCGGATCCTTGGTTTGCCTT	Forward 5′UTR *agd3* repair
FR300	TGCAGTGTCAGGGAACAGAG	Reverse 5′UTR *agd3* repair
FR301	CTTGTCACCCTCTCTGTTCCCTGACACTGCACCGCCCTTATGCGGGTCG	Forward 3′UTR *agd3* repair
FR302	GCTAAAGCTTAGCGTGGATC	Reverse 3′UTR *agd3* repair
CS14	AGTTCATATGAACACCGGCGTGGAGCAG	Forward *agd3*
CS37	TCTAGATCACAAAGCAATAGGAGTGGAAAGCG	Reverse *agd3*

* Restriction sites are in bold. Protospacer sequences are underlined.

## Data Availability

Data is contained within the article.

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
