# Peer review of "Identification of Compounds Preventing A. fumigatus Biofilm Formation by Inhibition of the Galactosaminogalactan Deacetylase Agd3"

_ijms, 2023, doi:10.3390/ijms24031851_

Round 1

Reviewer 1 Report

The manuscript presents the identification of inhibitors of GAG deacetylase, Agd3, in A. fumigatus, a fungus known for forming biofilms in the human airways that can create a physical barrier protecting it from chemicals such as antifungal drugs. This makes it difficult to eradicate A. fumigatus infections, and the development of novel therapeutic strategies that combine drugs inhibiting biofilm synthesis or disrupting existing biofilms with classical antimicrobials is an important area of research. The findings of this study suggest that targeting the enzyme Agd3 may be a promising approach for developing new therapies for A. fumigatus infections, making this work of broad interest to researchers studying deep-seated fungal infections and relevant to the development of antifungal drugs. The experiments were well-designed and the paper was well-written, with discussions covering relevant areas and supporting understanding of the results.

Minor points: It is stated that the addition of 100 μM rifaximin did not compromise biofilm formation. Was there any effect on biofilm formation when the concentration of rifaximin was increased?Why was an in-depth study on rifaximin not conducted?

There is a dearth of literature on the biosynthesis of galactomannan, with only self-referencing present.

Author Response

Dear Editor, Dear reviewers,

Thank you for your positive evaluation of our manuscript. We have addressed your concerns as described below.

Reviewer 1: It is stated that the addition of 100 μM rifaximin did not compromise biofilm formation. Was there any effect on biofilm formation when the concentration of rifaximin was increased? Why was an in-depth study on rifaximin not conducted?

We have concentrated our efforts on the most promising candidate and did not try higher concentration of rifaximin. At 100 µM, imatinib showed a clear effect on biofilm formation whereas rifaximin did not show any effect at all (as shown in fig. 4C). Rifaximin has also a rather low solubility in water (less than 1 mg/mL). I added the sentence “Due to its inability to disrupt the biofilm and its limited solubility in aqueous systems, rifaximin was not analyzed further.”  to clarify this point.  

There is a dearth of literature on the biosynthesis of galactomannan, with only self-referencing present.

You are perfectly right and I have corrected this mistake. To avoid an extensive list (since there are many publications on this subject), I have added two reviews by researchers that have extensively contributed to our knowledge of galactomannan structure and biosynthesis (see references 16 and 17). I also added a recent publication (reference 18) that is not addressed in these two reviews.

Reviewer 2: I believe the abstract could definitely be more informative – results should be briefly discussed in the abstract. In its current form there is nothing more than current state of art, in my opinion it would be beneficial to include brief insight into results.

The abstract has been revised to include the main results of this research project as requested.

Line 79 – Pichia pastoris – italics; Line 206 – Aspergillus fumigatus – italics; Line 249 – IcaB 

A native Australian speaker has edited the final text and corrected the underlined errors. I hope that the reading is now easier.

Sincerely,

Franҫoise Routier

Reviewer 2 Report

I believe the abstract could definitely be more informative – results should be briefly discussed in the abstract. In its current form there is nothing more than current state of art, in my opinion it would be beneficial to include brief insight into results.

Line 79 – Pichia pastoris – italics

Line 206 – Aspergillus fumigatus – italics

Line 249 – IcaB 

Author Response

Dear Editor, Dear reviewer,

Thank you for your positive evaluation of our manuscript. We have addressed your concerns as described below.

I believe the abstract could definitely be more informative – results should be briefly discussed in the abstract. In its current form there is nothing more than current state of art, in my opinion it would be beneficial to include brief insight into results.

The abstract has been revised to include the main results of this research project as requested.

Line 79 – Pichia pastoris – italics; Line 206 – Aspergillus fumigatus – italics; Line 249 – IcaB 

A native Australian speaker has edited the final text and corrected the underlined errors. I hope that the reading is now easier.

Sincerely,

Franҫoise Routier